# Comparing the Replication Fidelity of Solid Microneedles Using Injection Compression Moulding and Conventional Injection Moulding

**DOI:** 10.3390/mi13081280

**Published:** 2022-08-08

**Authors:** Tim Evens, Sylvie Castagne, David Seveno, Albert Van Bael

**Affiliations:** 1Department of Materials Engineering Diepenbeek Campus, KU Leuven, Wetenschapspark 27, 3590 Diepenbeek, Belgium; 2Department of Mechanical Engineering and Flanders Make@KU Leuven-MaPS, KU Leuven, Celestijnenlaan 300, 3001 Leuven, Belgium; 3Department of Materials Engineering, KU Leuven, Kasteelpark Arenberg 44, 3001 Leuven, Belgium

**Keywords:** microneedles, injection moulding, injection compression moulding, micro manufacturing, laser machining

## Abstract

Polymer surfaces are increasingly being functionalized with micro- and nano- surface features using mass replication methods such as injection moulding. An example of these are microneedle arrays, which contain needle-like microscopic structures, which facilitate drug or vaccine delivery in a minimally invasive way. In this study, the replication fidelity of two types of solid polycarbonate microneedles was investigated using injection compression moulding and conventional injection moulding. Using a full factorial design of experiments for the injection moulding process, it was found that the volumetric injection rate had the largest positive effect on the replication fidelity. The mould temperature and holding pressure were also found to have a positive effect, while the effect of the melt temperature was found to be insignificant for the considered temperature range. For the injection compression moulding process, it was found that a larger compression stroke resulted in a better replication fidelity. A comparison between the replication fidelity for the injection moulding and injection compression moulding indicated that the injection compression moulding process resulted in a higher and more uniform replication fidelity. Using finite element flow simulations, a higher and more evenly distributed cavity pressure was observed compared to the conventional injection moulding process.

## 1. Introduction

Integrating functional micro- and nano- surface features on polymeric surfaces has become increasingly popular over the last decades, as these products can offer a wide range of potential applications for different markets [1]. An example of this, found in the medical field are microneedle arrays, which are invasive needle-like micro-features, used to (i) deliver medical solutions or (ii) to extract fluids from a body for sampling applications. Various types of microneedles have been developed in the recent years, having various shapes and lengths ranging from 25 to 2500 µm [2]. Over the last years, we have developed a mass production method to produce solid [3] and hollow microneedle arrays [4]. In this method, polymer injection moulding is used to replicate microneedle cavities, which are created using laser ablation. The microneedle cavities can be laser ablated in various mould materials and the cavity geometry can be adapted by changing laser scanning parameters as shown in [5]. Moreover, in a recent work, we evaluated the simulated injection moulded replication fidelity using finite element flow simulations and found that the replication fidelity could be predicted with an accuracy between 94.5% and 97.0% [6].

Having a low tip radius is essential for the polymer microneedles, as it reduces the penetration force needed to pierce the skin [7]. Although the laser ablated microneedle cavities have already proven to be very sharp, it is still necessary to have a good replication fidelity during the injection moulding process. However, obtaining a high and uniform replication fidelity of micro surface features has proven to be challenging. In order to find injection moulding conditions with an optimal replication fidelity, various researchers have performed Designs of Experiments (DoE) [8,9,10,11,12,13,14,15,16,17,18,19,20,21,22]. As a result, it is possible to understand the effect of each injection moulding parameter on the replication fidelity. However, some researchers still observed an inhomogeneous replication fidelity over the product surface, due to an uneven temperature and pressure distributed in the mould cavity [12,15,23,24]. A possible candidate to overcome the inherent limitations of an unevenly distributed cavity pressure in conventional injection moulding, is injection compression moulding.

In injection compression moulding, the melt is injected into a slightly opened cavity, with the two mould halves initially being separated from each other [25]. After the injection phase (i.e., after filling 90–98% of the total product volume), the mould closes and compresses the melt in the cavity. The additional gap between the moulds is defined as the compression stroke, which provides the necessary distance to perform the compressing action. A major advantage of injection compression moulding is the ability to reduce stress in the part, as during the compression phase the closing force is applied on the entire product surface, ensuring a uniform pressure distribution inside the cavity [13,15,26]. For this reason, injection compression moulding has been extensively favoured to conventional injection moulding in the production of optical components, as it is proven to reduce birefringence (double refraction) and enhance optical properties [13,23]. However, the mould compression phase increases the complexity of the production process, and the additional compression stroke must be taken into consideration while optimizing the process. Suzuki et al. presented the importance of increasing the compression stroke in order to improve surface replication [27]. Yet on the contrary, Masato et al. [28] observed a negative impact of a large compression gap on the replication homogeneity for micro grooves with a depth of 15 µm and a width of 3 µm. Moreover, a significant interaction between the compression gap and the injection velocity was found, further confirming that the optimization of the compression gap must be taken into account in the overall polymer flow conditions. Injection compression moulding was also successively used to produce microlenses and periodic micro feature arrays by [29,30], yet the authors did not investigate the effect of the process parameters on the replication fidelity. Loaldi et al. compared the replication capability of injection moulding and injection compression moulding for a micro structured Fresnel lens. It was shown that the injection compression moulding improved the micro-replication while using a high holding pressure in combination with a lower compression stroke [13]. A few authors found that the mould temperature had a significant effect on the replication fidelity of micro- and nano- surface texture replicated using injection compression moulding, as high mould temperatures improved the replication fidelity [15,31,32].

Although most authors indicate the mould temperature to have a significant positive effect on the replication fidelity, the effect of the other injection moulding parameters is often inconclusive. In fact, some researchers found that the volumetric injection rate, holding pressure and melt temperature had a positive effect on the replication fidelity, while others found these parameters to have a negative or insignificant effect. Moreover, a wide variety of micro surface features have been investigated in the literature, yet most of them have a low aspect ratio (<2) with geometrical dimensions in the range of 10–100 µm. A detailed study on the effect of processing parameters in injection moulding for the replication of large micro surface features with high aspect ratios is currently lacking. The investigated microscopic products in the literature are usually small flat plates in the order of a few millimetres, yet the size of the product surface will play an important role in the replication fidelity and replication uniformity. This replication uniformity over the complete product surface is often overlooked or poorly discussed. Although injection compression moulding is often applied to manufacture polymeric products with micro- and nano- surface features, no studies have been found which compare the replication fidelity of large micro-features for conventional injection moulding and injection compression moulding, while taking into account the difference in flow behaviour.

In this study, we investigate the replication fidelity of solid polycarbonate microneedles using injection compression moulding and conventional injection moulding. For this, a mould cavity insert for a large 112 mm × 70 mm × 1.7 mm flat polymer plate is created. Two types of microneedle cavities having a depth of 1959 µm and 1125 µm are laser ablated on 15 locations. A DoE will be performed, to define process parameters which correspond to an optimal replication fidelity for the microneedles in the injection moulding process. Furthermore, the compression stroke, which correlates with an optimal replication fidelity of the microneedles will be determined for the injection compression moulding process. Afterwards, a comparison between the replication fidelity of the injection moulding and injection compression moulding process will be made, and the difference in filling behaviour will be discussed using finite element flow simulations.

## 2. Materials and Methods

### 2.1. Thermoplastic and Mould Material

The thermoplastic material used in this study is polycarbonate (PC, Lexan™ HPX8REU, manufactured by Sabic, Bergen op Zoom, The Netherlands), as this grade is suitable for medical devices and pharmaceutical applications. The thermoplastic has a high flowability indicated by the melt mass-flow rate of 35 g/10 min (300 °C/1.2 kg) and a glass transition temperature of 141 °C.

The selected material for the mould insert is a low corrosion tool steel “Stavax” (grade 1.2083—AISI 420), having a high corrosion and wear resistance.

### 2.2. Mould Product Cavity and Laser Machining Experiments

The mould product cavity is a 112 mm × 70 mm × 1.7 mm flat plate, which is machined within the tool steel mould insert. Moreover, two different arrays of microneedle cavities were machined in the mould product cavity on 15 locations, as illustrated in Figure 1. The machining of the microneedle cavities was performed using a femtosecond laser ablation process, as described in a previous work [3]. In this strategy, the laser spot scans a circular region with a specified diameter and follows parallel lines in two perpendicular directions. The distance between two consecutive lines is defined as the hatch pitch. Once the laser has scanned one circular grid, the focal point is lowered (with a vertical distance defined as the layer pitch) and the laser scans again the same area. This process is repeated multiple times for a prescribed number of layers. The hatch pitch and layer pitch were fixed at 15 μm and 2 μm, respectively. The first type of microneedle cavities will be referred to as “large microneedles” and were created using a base diameter of 400 µm and 600 layers. The second type of cavities will be referred to as “small microneedles” and were created using a base diameter of 300 µm and 150 layers.

### 2.3. Experimental Injection Moulding Set-Up

The injection moulding machine used for this study is an Engel ES 200/35 HL hydraulic injection moulding machine with a maximum clamping force of 350 kN and a 25 mm horizontal screw with an L/D ratio of 24.8. The machine is equipped with a hydraulic accumulator to increase the volumetric injection rate and the injection speed to 149 cm^3^/s and 300 mm/s, respectively. Moreover, the injection moulding machine is also equipped with an injection compression moulding module, which allows the machine to be used for both conventional injection moulding (IM) and injection compression moulding (ICM). The ICM process allows for the polymer melt to be injected in a slightly open mould with a defined compression stroke. Once the injection phase has reached the switch-over point, the machine applies the clamping force and closes the two mould halves. The temperature of the mould is controlled by a Wittmann Tempro D controller. The polymer is injected through a hot runner, located in the centre of the mould product cavity, as illustrated in Figure 1.

The stationary side of the injection mould is equipped with a temperature sensor (Priamus, 4008B, Priamus System Technologies, Schaffhausen, Switzerland) and a piezoelectric pressure sensor (Priamus, 6001B), which are located opposite to array number 7. The sensors allow for precise process control and online measurement of the apparent viscosity. This viscosity measurement is a built-in module in the Priamus FILLCONTROL. The online apparent viscosity is measured for the two volumetric injection rates used in this study, being 50 cm^3^/s and 150 cm^3^/s.

The barrel temperature, volumetric injection rate, holding pressure and mould coolant temperature were varied in the DoE, while the holding pressure time and cooling time were kept constant through all experiments for both IM and ICM at 10 s and 20 s, respectively. The additional ‘compression stroke’ parameter within the ICM process was varied from 1 mm to 5 mm in steps of 1 mm, at a compression speed of approximately 50 mm/s. For each experimental run, the first 20 moulded parts were discarded, and the following 15 samples were kept for evaluation.

### 2.4. Two-Level Full Factorial Design

A two-level full factorial design was formulated to identify the combination of injection moulding parameters that have the highest replication fidelity during IM. With a 2^4^ design it is possible to analyse both the main effect as well as the two- and three-factor interactions [33]. Parameters varied in this study are the barrel temperature, volumetric injection rate, holding pressure and mould coolant temperature, as these parameters are widely reported in the literature as the most influential for the replication fidelity of micro-features. The levels of the process parameters were identified in preliminary experiments, starting from the manufacturer’s recommendations and aiming at obtaining a consistent filling without the occurrence of needle breakage, short shots, sink marks or flash. The different settings for the production of the samples are presented in Table 1. Each setting was repeated 3 times and the injection moulding experiments were conducted in a random order. For each injection moulded sample, the average replicated microneedle length was calculated for both types of microneedles. To account for possible replication differences within the array, the first needle on the first row, the second needle on the second row and the third needle on the third row were measured for all arrays. The length of the replicated microneedles was used as input data in the statistical software Minitab 17. The significance of each of the resulting terms was tested using a *t*-test with 0.050 as the significance level. The null hypothesis for this test is that the effect of a specific term is zero. When the *P*-value for a specific term is smaller than 0.050, we can reject the null hypothesis, concluding that the specific term is significant.

### 2.5. Topography Characterisation

The geometries of the ablated microneedle cavities were characterized using a Phoenix Nanotom micro-computed tomography (μ-CT) system. The device is equipped with a high-power nanofocus X-ray tube and a diamond-tungsten target was chosen for the high X-ray absorbing metal samples. For each CT-scan, 2400 X-ray 2D projection images were obtained from incremental rotation of the scanned samples over 360°. Acquisition parameters were fixed for all samples as follows: voltage = 100 kV, current = 158 A, voxel size = 3.75 μm, and a 0.1 mm copper and 0.1 mm aluminium filter were used during scanning. Reconstructed XY datasets (slices) were exported from the software for further analysis and visualization within Fiji ImageJ. Before conducting measurements, the Nanotom instrument was calibrated using a calibration rod (Goekeler Messtechnik, VTX18CE000-022, Lenningen, Germany) with an uncertainty of 1.0 µm. For both types of microneedles, a 3 × 3 array of cavities was laser ablated on a tool steel sample (grade 1.2083—AISI 420), with identical laser scanning parameters to the cavities produced on the mould insert. The average length and diameter of these 9 holes along with a 95% confidence interval, using the Student’s t-distribution was reported in the plotted results.

The geometries of the replicated thermoplastic microneedles were characterized using a digital microscope (model Keyence VH-S30, Keyence, Osaka, Japan) with a maximum magnification of ×200. The system is connected to a VHX-500 F monitor with built-in measuring software. The microscope was calibrated using a stage micrometre (Olympus Tokyo, OBMM 1/100, Tokyo, Japan) having an uncertainty of 0.5 μm.

### 2.6. Injection Moulding Simulations

The commercial software package Moldex3D studio (version 2021, Zhubei, Taiwan) was used to simulate the injection moulding process. In a previous study, the simulated replication fidelity of polymer microneedles using Moldex3D was compared to injection moulding experiments [6]. It was found that the replication fidelity of the microneedles could be predicted with an accuracy between 94.5% and 97.0%, using multi-scale meshing and optimizing the Heat Transfer Coefficient (HTC). An ICM module was also tested in the current study to understand the behaviour of the melt flow inside the macroscopic part. Predicting the replication fidelity inside the microneedles, however, was found to be unreliable as mesh elements were often filled inside the microneedle tip, without being in contact with already filled elements. Therefore, the ICM module was only used for simulating the macroscopic flow, while it was not used to predict the microscopic replication fidelity. All simulations in this study were conducted using an Intel Core i7-9850H Processor with 12 cores and a random-access memory (RAM) of 16 GB.

The surface of the polymer part was meshed using a 5-layer prismatic boundary layer mesh. Inwards from this surface mesh, a tetrahedral mesh filled up the remaining space, to obtain a total 11 layers across the part thickness. This number of layers ensures an adequate representation of the laminar melt flow. Multi-scale meshing was used, in which the mesh is refined within the microfeatures to achieve a precise simulation, while a coarser mesh is used in the macroscopic part. In our previous study [6], a mesh convergence study was conducted to define a satisfactory mesh size. It was found that the replication fidelity of the microneedles converged around 2.4 million mesh elements, corresponding to a mesh edge length of approximately 100 µm in the large microneedles and 50 µm in the smaller microneedles. Therefore, in this study, a mesh edge length of 100 µm was also used in the large microneedles and 50 µm in the small microneedles, resulting in a total number of approximately 3.9 million solid mesh elements.

The injection moulding parameters, being the holding pressure, melt temperature and mould coolant temperature were set to the same values as used in the experiments. The injection speed in Moldex3D was defined as a filling time, which was calculated using the volume of the part and the known volumetric injection rate. The calculated time corresponded to 0.1 s, which is in agreement with the experimental filling time of the injection moulding machine. Another important model parameter is the HTC, which defines the heat transfer between the polymer melt and the surrounding mould material. In our previous study [6], we determined the HTC value, having the best correspondence with the experimentally determined replication fidelity, by manually adjusting it. For PC in combination with a tool steel insert, an optimal value of 30,000 W/m^2^∙K was found. The HTC during the packing and detached phase were set to the default values used in Autodesk Moldflow, being 2500 W/m^2^∙K and 1250 W/m^2^∙K, respectively. The last parameter that was adapted from the default value is the criterion for stopping the calculation. This parameter is linked to a function which aims to decrease the computational time. By default, this value is set as 99.95%, which means that if the product is filled for 99.95%, Moldex3D assumes that the part is completely filled, without a short shot and will end the simulation early to save computational time. In our case, the incomplete filling of the microfeatures is smaller than 0.05% of the complete volume of the part, thus Moldex3D displays the part as completely filled. To overcome this problem, the criterion for stopping the calculation is changed into 99.9999%, enabling the simulation of incompletely filled microfeatures. A summary of the simulation parameters used in Moldex3D can be found in Table 2.

## 3. Results and Discussion

### 3.1. Geometry of the Ablated Microneedle Cavities

Figure 2 represents the XY datasets of the reconstructed µCT analysis for three large and three small microneedle cavities. The average cavity depth and cavity base diameter, along with the standard deviation for the complete array of nine microneedles are also displayed for both types of microneedles. A clear difference in dimensions and shape is observed between both type of microneedle cavities, due to the difference in laser scanning parameters used for both types of microneedles. The effect of these laser scanning parameters—the programmed base diameters and number of layers—on the cavity shape and dimensions are discussed in a previous study [5].

Although both types of microneedle cavities are different in shape and dimensions, it is clear that they both have very sharp tip radii. This is essential for creating sharp polymer microneedles. Yet, completely filling these needle tips will be nearly impossible in practice. However, the part of the needle tip that cannot be filled will act as a kind of buffer in which entrapped air can be collected during the filling process, thus reducing the risk of the so-called “diesel effect”. This effect is an undesirable phenomenon causing thermal degradation of the thermoplastic, due to a very high pressure and temperature resulting from highly compressed air.

### 3.2. DoE to Define IM Parameters for a High Replication Fidelity

The influences of the barrel temperature, volumetric injection rate, holding pressure and mould coolant temperature on the replication fidelity of two types of injection moulded microneedles were assessed by a full factorial DoE. During the injection moulding experiments, both microneedle arrays located in the middle of the part (array 8 on Figure 1) broke during demoulding, leaving the microneedles stuck inside the cavities. The reason why these microneedles broke was at first sight unclear, and array 8 was therefore excluded from this study. To understand why the needles from array 8 failed, we created a cross section of the microneedle cavities using a metallographic saw and a high precision surface grinder. Figure 3 illustrates the grinded cross section of an empty large microneedle cavity and a large microneedle cavity from array 8 with an entrapped broken microneedle inside. It can be observed that the polymer almost completely filled the microneedle cavity. It is expected that the micro- and nano-surface roughness inside the microneedle cavities, which are induced during the laser ablation process, are therefore also replicated. This replicated roughness causes microscopic undercuts on the microneedle shaft, which result in the needle breakage during demoulding. Hopmann et al. also observed high demoulding forces, which caused micro-structures to break during demoulding [34]. The reason why the replication fidelity is so high for the microneedles from array 8, is due to the cavities being located directly opposite to the gate location. From the injection moulding simulations, it is observed that the polymer is directly injected into the microneedle cavities, causing already a high replication fidelity during the initial filling stage. Moreover, the pressure inside the microneedle cavities is very high during the packing phase, as the cavities are located very close to the gate location.

The factorial regression analysis assumes that the observations are normally distributed, randomised and of equal variance at all response levels. By using the analysis of residuals, these three requirements were verified for both types of microneedles [33]. Figure 4 reports the average replication in height for the 16 DoE runs with a repetition of three for both the (a) large microneedle arrays and (b) the small microneedle arrays. No significant difference was noticeable between the three repetitions due to the overlapping confidence intervals, thus indicating a good repeatability of the injection moulding process. Clear differences in replication fidelity can be observed between the different DoE runs. This is further illustrated in Figure 5, which depicts one replicated large and small microneedle from array 2 for DoE run 3, 11, 12 and 13.

The results of the DoE analysis for both the large and small microneedles are depicted in Figure 6. The Pareto charts, main effect plots and two-factor interactions are shown. First, the four main effects are discussed for the large microneedles. From the Pareto chart, it can be observed that all main effects have a significant effect on the replication fidelity of the microneedles. The volumetric injection rate has by far the largest positive effect on the replication in height, ranging from approximately 600 to 1200 µm for respectively, 50 and 150 cm^3^/s. This parameter is followed by the holding pressure and the mould coolant temperature, while the melt temperature has the lowest effect on the replication fidelity of the microneedles. Similar results can be found for the small microneedles, yet for these smaller features it can be observed that the melt temperature has no significant effect on the replication fidelity. The volumetric injection rate is once again the key parameter affecting the replication fidelity, ranging from approximately 300 to 650 µm for respectively, 50 and 150 cm^3^/s. The mould and holding pressure have an almost identical positive effect on the replication fidelity.

From the two-factor interaction plots it is clear that for the large needles, the combination of holding pressure and mould temperature had a positive effect on the replication fidelity, while for the small needles this effect was insignificant. A possible explanation for this is that due to the higher mould temperature, the large needles were not completely solidified yet and the higher holding pressure further increased the degree of filling during the packing phase. This effect was not significant for the smaller needles, as these features solidify much faster due to their lower volume and did not deform during the packing phase. The volumetric injection speed in combination with the mould temperature and the volumetric injection speed in combination with the barrel temperature had a significant positive effect on the replication fidelity for both types of microneedles. For both cases, it can be assumed that a high volumetric injection speed in combination with a higher mould or barrel temperature increased the temperature of the polymer melt and decreased the melt viscosity, resulting in a higher replication fidelity.

For both types of microneedles, all main parameters and two-factor interactions were found to have a positive effect on the replication fidelity. This facilitates the setup of the injection moulding process, as each injection moulding parameter should be set to the highest achievable value without causing injection moulding defects on the products.

#### 3.2.1. Volumetric Injection Rate

The volumetric injection rate has, in both types of microneedles, the highest positive effect on the replication fidelity. In fact, the volumetric injection rate has two positive effects on the polymer melt that improve the replication fidelity. First, an increase in the volumetric injection rate will promote shear heating, which will locally decrease the melt viscosity. Second, a high volumetric injection rate reduces the injection time, which in turn delays the formation of the frozen skin layer [1,35]. From the literature, this parameter is known to have a positive effect on the filling of micro and nano surface features, as described by several researchers [10,11,12,13,14,36,37,38,39,40,41,42]. However, high volumetric injection rates have also been reported to cause polymer degradation due to exceeding the maximum shear stress allowed by the polymer. Moreover, the high generated shear can also cause internal stress in the part, which could lead to damage of the micro-features or a poor surface quality [35,36]. 

Figure 7 shows both the Cross–WLF (Cross–Williams–Landel–Ferry) viscosity curves for the investigated PC grade, extracted from Moldex3D and the online apparent viscosity, measured during injection moulding experiments using a barrel temperature of 315 °C. A shear rate of approximately 500 1/s and an apparent viscosity of 106 Pa.s was found for a volumetric injection rate of 50 cm^3^/s. While for the volumetric injection rate of 150 cm^3^/s a shear rate of 9900 1/s and an apparent viscosity of 30 Pa.s was found. This significant reduction in viscosity caused by the increase in volumetric injection rate, has for sure a positive effect on the replication fidelity during the filling stage.

#### 3.2.2. Holding Pressure

The second most important parameter influencing the replication fidelity is the holding pressure. In conventional injection moulding, this holding pressure compensates the polymer shrinkage to achieve the desired geometry after demoulding. In the injection moulding of micro and nano surface features, however, the holding pressure plays an additional important role in the filling of the micro-features. The holding pressure is the key parameter that can force the polymer melt into the micro and nano cavities [1,43]. No matter how high the melt temperature or how low the melt viscosity, a high pressure is still required to achieve a good replication fidelity. The positive effect of holding pressure on the replication quality of micro-features is also shown in multiple studies [8,10,13,17,18,19,37].

#### 3.2.3. Mould Temperature

The mould temperature was found to have a significant positive effect on the replication fidelity of the large and small microneedles. In the literature, the mould temperature is generally considered to be the most important parameter when it comes to replicating micro-features [8,9,10,11,15,18,19,22,37]. During the filling stage, the filling of the micro-features has to compete with the filling of the rest of the cavity. Yet, most of the material will flow in the main direction of the macroscale cavity. The polymer will follow the path of least resistance, leaving the micro-features only partially filled, also known as the hesitation effect [28]. Further filling the micro-features will only be possible during the packing stage, as during this phase the pressure in the cavity is substantially increased. However, it is impossible to further increase the filling of these micro-features, if they are already solidified at this moment. Therefore, it is essential that the solidification of the skin layer is delayed so the polymer can still be deformed during the packing phase. Some authors even implemented a rapid heat cycling approach, also called variothermal injection moulding (VIM,) to further improve the replication fidelity [12,17,20,21,31]. Here, the mould temperature is no longer a constant, but it is varied between a high value during the filling stage and a lower value during the cooling stage. The high value during the filling stage is typically taken above the glass transition temperature for amorphous materials and around the melting temperature for semi-crystalline polymers. This way, the formation of a frozen layer is delayed, hence leading to a better filling of micro-features. VIM is more complex and more expensive (due to more expensive moulds and side equipment) than IM. In addition, cycle times are substantially higher compared to conventional injection moulding. Mainly due to the increase in cycle time, this system was not considered in this study.

In this study, the mould temperature was found to have a significant positive effect on the replication fidelity, yet it is not the parameter with the highest effect. This interesting difference compared to several other studies, is expected to be due to two factors. First, there is a difference in size of the investigated micro-features compared to typical micro-features found in the literature. In our study, the micro-features are microneedle cavities with a depth of approximately 1 mm and 2 mm, having cavity diameters of approximately 0.25 mm and 0.5 mm for the small and large microneedles, respectively. The investigated microcavities found in the literature, however, are typically a few tens of micrometres in depth and width. The small microfeatures found in the literature have a much higher surface area to volume ratio, which results in a much higher heat transfer between the molten polymer and the mould surface, which in turn results in a faster solidification of the polymer. Thus, for these smaller microfeatures it is crucial to delay the solidification by increasing the mould temperature. Secondly, some authors used the rapid heat cycling approach to increase the mould temperature, even above the glass transition temperature for amorphous polymers, before the injection phase. By doing so, the replication fidelity was significantly improved compared to conventional isothermal injection moulding. Therefore, several authors found the mould temperature to be the parameter with the highest effect on the replication fidelity.

#### 3.2.4. Melt Temperature

The last investigated parameter is the melt temperature. This parameter only had a small impact on the replication fidelity of the large microneedles and was even insignificant for the small microneedles. In the literature, this parameter is also only found to have an impact on the replication fidelity of micro-features in a few studies [8,17,19]. A high melt temperature will decrease the melt viscosity, which has a positive effect on the replication fidelity of micro-features. A melt temperature above the recommended temperature range of the polymer manufacturer, however, can induce thermal degradation of the polymer inside the barrel and should therefore be avoided.

### 3.3. IM Using Parameters for an Optimal Replication Fidelity

The DoE indicated that increasing all of the investigated parameters, being the barrel temperature, volumetric injection rate, holding pressure and mould temperature had a positive effect on the replication fidelity of the microneedles. Therefore, when aiming for an optimal replication fidelity, each parameter should be set to the highest possible value without causing injection moulding defects. The barrel temperature is set to the highest recommended value provided by the polymer manufacturer, being 315 °C. Increasing the temperature beyond this value could lead to thermal degradation of the polymer. The volumetric injection rate is set to the highest achievable value limited by the injection moulding machine, in this case 150 cm^3^/s. The holding pressure was varied throughout injection moulding experiments to find the highest possible pressure without causing injection moulding defects. A maximum holding pressure of 63 MPa was found, as flash appeared above this value. The mould coolant temperature was also varied throughout injection moulding experiments. Starting at 115 °C, the mould coolant temperature was increased in steps of 5 °C. At a coolant temperature of 130 °C, which is still below the glass transition temperature of 141 °C, some needles deformed or even broke during the demoulding process. Microscopic images of an undeformed and deformed small microneedle are shown in Figure 8. Even when extending the cooling time to 30 s, these defects occurred. Masato et al. also observed breakage of micro-features during demoulding at very high mould temperatures [28]. It is expected that similar to the microneedles from array 8, the replication fidelity is very high, which causes the micro- and nano-surface roughness inside the microneedle cavities to be replicated, this time due to the high mould temperatures. Moreover, it should be noted that the tensile strength of the polymer material is lower at higher temperatures. Thus, it is expected that at high mould temperatures, the needle deformation and breakage are the result of higher demoulding forces due to a higher replication fidelity and a lower tensile strength of the polymeric material. Thus, the optimal replication fidelity is not defined by the best filling of the microcavities but by the best trade-off between filling and demoulding forces. To ensure that no defects occur, the maximum mould coolant temperature is set at 125 °C. A summary of the IM parameters for an optimal replication fidelity are shown in Table 3.

Figure 9 shows the experimental and simulated replication in height of the small and large microneedle arrays created with the IM parameters for an optimal replication fidelity. A good correlation between the experimental and simulated replication in height is observed both for the large and small microneedles with a percentual absolute average deviation of 4.4% and 5.1%, respectively. The average replication in height together with the standard deviation for the large and small microneedles are 1371 ± 107 µm and 681 ± 41 µm, respectively. This corresponds to an average replication fidelity of 70.0% for the large microneedles and 60.6% for the small microneedles.

Figure 9 clearly shows that there is a deviation in the replication fidelity across the different microneedle arrays. The cause of this deviation can be allocated to the different positions of the microneedles on the macroscopic part, which cause a difference in temperature, pressure and viscosity of the polymer melt during the filling of the microneedle arrays. In fact, when the replication fidelity of the microneedle arrays is displayed as a function of the distance between the corresponding microneedles and the gate location, a clear trend is observed, as shown in Figure 10. A strong linear relationship is found between the replication in height and the location of the microneedles from the gate, indicated by the high values of the linear correlation coefficient (R). This trend is also observed within the injection moulding simulations, as shown in Figure 10b, where the experimental and simulated replication fidelity is shown as a function of the distance from the gate. Explaining this trend was recently completed within our latest paper, in which the replication fidelity of the microneedle close to the gate and far from the gate were compared [6]. Here, it was found that the microneedles close to the gate were partially filled during the initial filling phase. Yet, when the macroscopic part was completely filled and the pressure in the cavity increased, the temperature of the melt inside the microneedles had already decreased below the no-flow temperature, thus preventing further filling of the microneedle cavities. The filling of the microneedles located far from the gate, however, was delayed due to the increased flow length. During the filling of these microcavities, the temperature of the melt was still very high due to the characteristic fountain flow present in injection moulding. Only shortly after the melt reached the microcavities far from the gate, the macroscopic part was completely filled, and the pressure reached a maximum. At that moment, the temperature within the cavities located far from the gate was still above the no-flow temperature and thus ensured complete filling. This behaviour is also present in the current study, as the farther the microfeatures are located from the injection location, the higher the melt temperature is at the moment when the cavity pressure rises. Thus, microneedles located far from the gate have a higher replication fidelity compared to microneedles located close to the gate.

### 3.4. Injection Compression Moulding

The defined injection moulding parameters for an optimal replication fidelity of the microneedles were also used during the injection compression moulding experiments. Therefore, the additional parameter in the ICM process, being the compression stroke, was varied in five steps. Figure 11 illustrates the average replication in height for both the large and small microneedle arrays, in function of the compression stroke. A clear trend can be observed. The average replication in height increases with an increase in the compression stroke and stabilizes around 4 mm. This trend can be explained by Figure 12, where for the different compression strokes, the injected melt volume before the mould compression phase is shown inside the mould cavity. The contact area between the injected polymer melt and the mould wall decreases with an increase in the compression stroke. If polymer material is in contact with the mould surface, it will rapidly solidify and form a frozen layer, due to the temperature difference. For the smaller compression strokes, a higher area of polymer will be in contact with the mould, therefore a large area of frozen layer is formed, which will hinder the complete replication of the microneedles during the compression phase. For the larger compression strokes, however, less material is in contact with the mould, therefore the bulk of the polymer melt is still at a high temperature after the initial injection phase. Afterwards, when the mould compression is applied, the bulk of the polymer melt is compressed and distributed over the mould surface, while still having a high temperature. Therefore, the polymer material can still be deformed during the compression phase, which results in a higher replication fidelity of the microneedles.

### 3.5. Comparison between IM and ICM

Figure 13 shows the average replication in height of the microneedle arrays as a function of the distance between the corresponding array and the gate location, for injection moulding and injection compression moulding using the process parameters for an optimal replication fidelity. A difference in replication fidelity can be observed between the microneedles produced with IM and ICM. The average replication in height together with the standard deviation, for all large microneedle arrays correspond to 1536 ± 81 µm and 1371 ± 117 µm for ICM and IM, respectively. For the small microneedle arrays, an average replication in height together with the standard deviation of 775 ± 38 and 675 ± 46 was found for ICM and IM, respectively. This corresponds to an average replication fidelity for the large microneedle arrays of 79% for ICM and 70% for IM. For the small microneedles, a replication fidelity of 68% and 61% was found for ICM and IM, respectively. Thus, for both the large and small microneedle arrays, a higher replication fidelity was found for ICM compared to IM. Moreover, a better replication uniformity across the different arrays was achieved for the ICM process, as indicated by the lower standard deviation. Microscopic images of a large and small microneedle from array 1, created with the injection moulding conditions for most optimal replication fidelity, being ICM with a mould stroke of 4 mm, are displayed in Figure 14. The cause of difference in replication fidelity between the two techniques will be discussed in the following section.

Figure 15 illustrates the simulated melt front time, temperature and pressure in the macroscopic part and a small microneedle from array 1, for IM and ICM with a stroke of 4 mm, at a filling time of 0.1 and 0.2 s. For IM, a filling time of 0.1 s corresponds to the switchover position where the filling changes from velocity driven to pressure driven. For ICM, this time corresponds to the compression switch, at which the mould will compress the polymer melt using the clamping force. Hence, a different filling behaviour is observed after 0.1 s between both injection moulding techniques. For the IM process, the polymer melt has filled the product cavity for approximately 98% and the microneedles are partially filled due to the pressure inside the mould cavity. For the ICM process, however, the complete product volume is injected into the enlarged mould cavity, creating a bulk of material having only a small contact area with the mould. Moreover, nearly no pressure is present in the cavity, resulting in almost no filling inside the microneedle cavities. At a filling time of 0.15 s, the macroscopic product is completely filled in the IM process, due to the additional pressure-controlled filling. At that moment, the pressure inside the mould cavity is increased to approximately 35 MPa, yet the partially filled microneedles are already cooled down and solidified, which means that the increased pressure cannot improve the filling any further. Moreover, it should be noted that this pressure is generated at the melt entrance and has to be distributed through the polymer material, resulting in an uneven pressure distribution. In the case of the ICM process, the complete melt volume is dispersed in the mould cavity, due to the closing stroke of the mould, at a filling time of 0.2 s. Although the complete filling takes more time compared to the IM process, the microneedles have a higher replication fidelity. This is due to the fact that the microneedle cavities were not yet filled at the initial filling stage of 0.1 s, and the bulk of material was not in contact with the mould surface, therefore retaining more heat. Moreover, a higher cavity pressure was observed in the ICM process of approximately 50 MPa, which is equivalent to the applied closing force of 350 kN on the product surface. Thus, due to the higher temperature of the melt inside the microneedle cavities, combined with a higher and more evenly distributed pressure inside the cavity, a higher replication fidelity with a better replication uniformity was found for the ICM process.

## 4. Conclusions

The present paper investigated the replication fidelity of solid polycarbonate microneedles using injection compression moulding and conventional injection moulding. For this, a mould cavity insert for a flat polymer plate was developed, on which two types of microneedle cavities were laser ablated on 15 locations. The first type of cavities, corresponding to large microneedles, had an average cavity depth of 1959 µm with a cavity base diameter of 430 µm, while the second type of cavity, corresponding the small microneedles, had an average cavity depth of 1125 µm with a cavity base diameter of 246 µm. A two-level full factorial design of experiments was conducted to define injection moulding parameters for an optimal replication fidelity. The investigated parameters were the barrel temperature, volumetric injection rate, holding pressure and mould coolant temperature. For the large microneedles, it was found that all parameters had a significant positive effect on the replication fidelity, with the volumetric injection rate having the largest positive effect. Moreover, all first order interactions, although not all significant, were found to have a positive effect on the replication fidelity. Similar results were found for the small microneedles, yet here the barrel temperature was found to be insignificant. The reason why the volumetric injection rate had the largest positive effect, was first due to the higher shear heating which decreased the melt viscosity and secondly due to a reduction in the injection time, which in turn delayed the formation of the frozen skin layer.

Finite element flow simulations were used to discuss the filling behaviour of the injection moulding process. A good correlation between the experimental and simulated replication fidelity was observed both for the large and small microneedles with a percentual absolute average deviation of 4.4% and 5.1%, respectively. Moreover, a deviation in the replication fidelity across the different microneedle arrays was observed and explained using flow simulations. The deviation in replication fidelity between the different microneedle arrays could be allocated to a different temperature, pressure and viscosity of the melt during the filling of the different needles.

The injection stroke within the injection compression moulding process was varied, and the effect on the replication fidelity was assessed. It was found that a larger compression stroke resulted in a better replication fidelity, with a stagnation after a stroke of 4 mm. Due to the larger compression stroke, less polymer material was in contact with the mould surface during the initial filling stage, which resulted in less heat transfer between the molten polymer and the mould surface.

A comparison between the replication fidelity of the injection moulding and injection compression moulding process was made, using the established parameters for an optimal replication fidelity. It was found that the injection compression moulding process resulted in a higher average replication fidelity together with a better replication uniformity across the different microneedle arrays, compared to the injection moulding process. The average replication in height for the two types of microneedles corresponded to from 79% and 68%, respectively. The cause of the difference in replication fidelity between both techniques was allocated to the different flow behaviours, which was discussed using finite element flow simulations. As the polymer was injected into a slightly opened mould for the injection compression moulding process, more heat was retained in the polymer melt, which promoted the filling of the microneedles. Moreover, a higher and more evenly distributed pressure inside the cavity was observed compared to the injection moulding process, due to the evenly distributed closing force on the product.

Thanks to this work, we better understand the effect of the injection moulding parameters on the replication fidelity of the microneedles. Moreover, we now know it is preferential to use injection compression moulding to produce solid polymer microneedles in the future. 

Throughout the study we observed some needle deformation and needle breakage, possibly due to a high demoulding force of the microneedles inside the cavities. This excessive demoulding force was caused by the replication of surface roughness on the microneedle shaft, which formed microscopic undercuts. In a future work, we will try to micro-polish the microcavities (using laser ablation or chemical etching techniques) and apply PVD coatings inside the microneedle cavities to reduce the surface roughness inside the microcavities.

## Figures and Tables

**Figure 1 micromachines-13-01280-f001:**
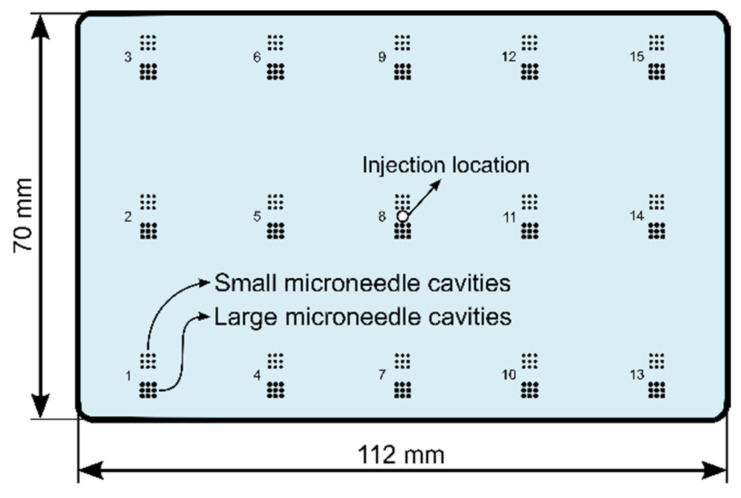
Illustration of the flat plate mould product cavity, including the two different arrays of microneedle cavities located on 15 locations. The polymer is injected through a hot runner, located in the centre of the mould product cavity.

**Figure 2 micromachines-13-01280-f002:**
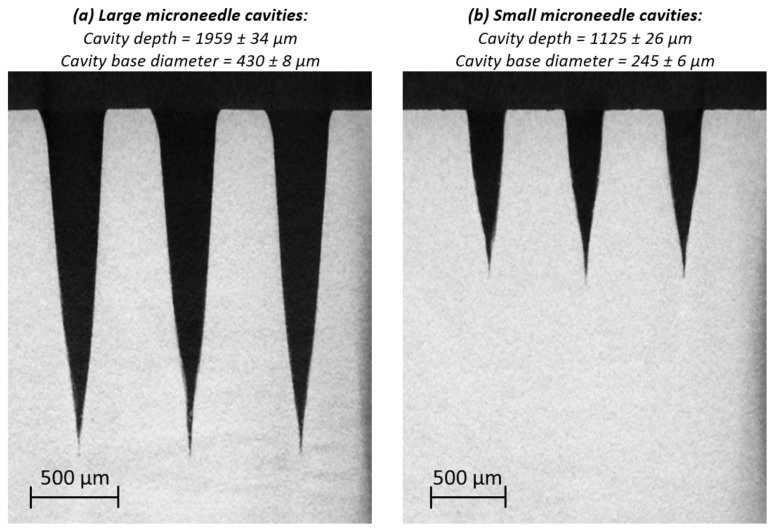
Representation of the XY datasets of the reconstructed µCT analysis for 3 large and 3 small microneedle cavities. The average cavity depth and cavity base diameter, along with the standard deviation for the complete array of 9 microneedles is also displayed for both types of microneedles.

**Figure 3 micromachines-13-01280-f003:**
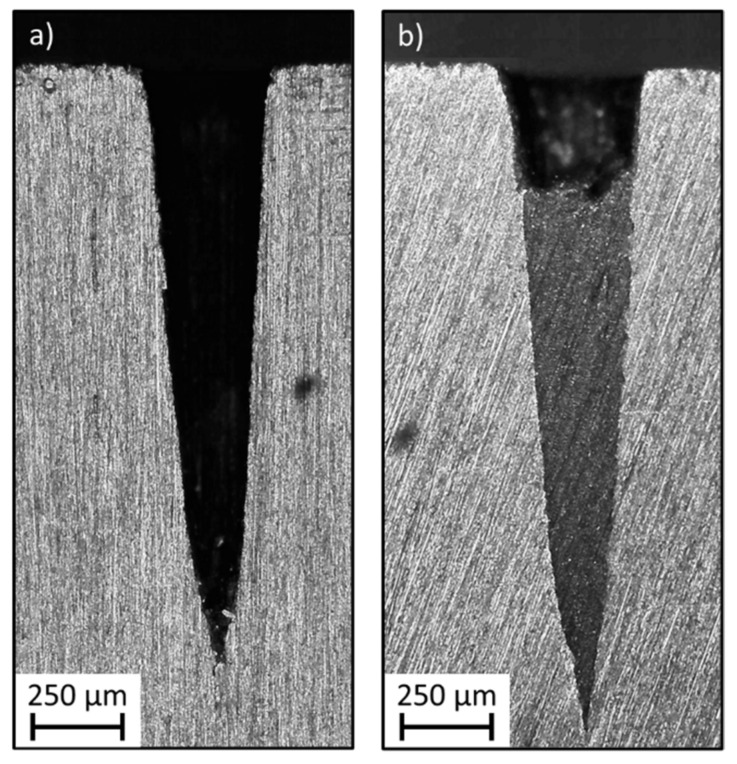
Illustration of a cross section of a microneedle cavity for (**a**) an empty large microneedle cavity, and (**b**) a large microneedle cavity from array 8 with an entrapped broken microneedle.

**Figure 4 micromachines-13-01280-f004:**
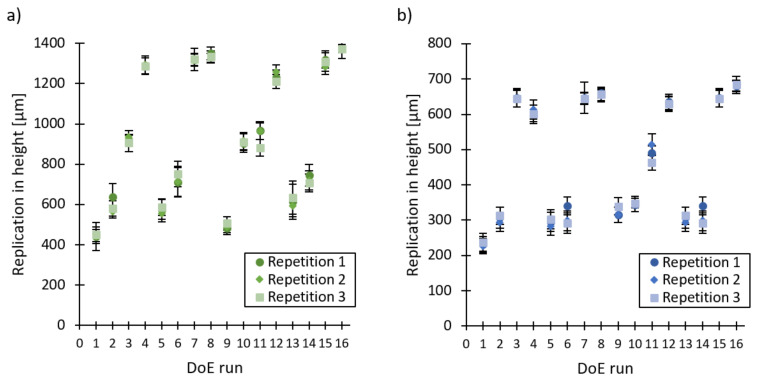
Average replication in height plotted against the 16 DoE runs with a repetition of 3 for (**a**) the large microneedle arrays and (**b**) the small microneedle arrays. The error bars represent the 95% confidence interval.

**Figure 5 micromachines-13-01280-f005:**
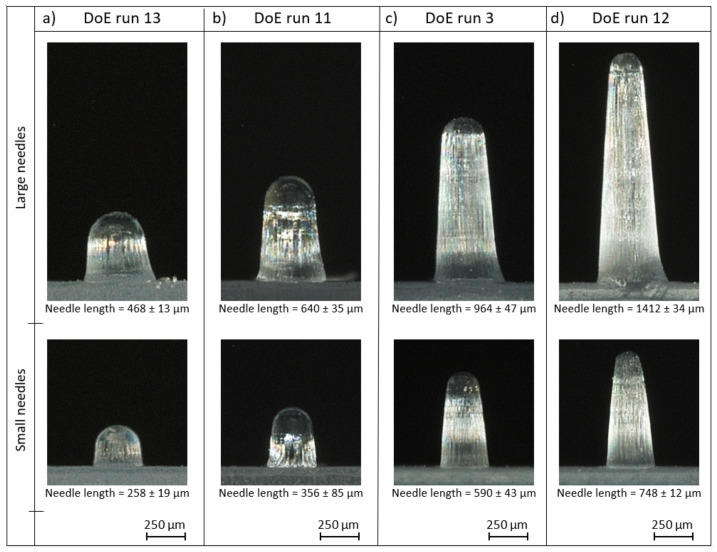
Illustration of one replicated large and small microneedle from array number 2 for (**a**) DoE run 13 (**b**) DoE run 11, (**c**) DoE run 3 and (**d**) DoE run 12. The average length of the needles is reported together with the standard deviation (*n* = 9).

**Figure 6 micromachines-13-01280-f006:**
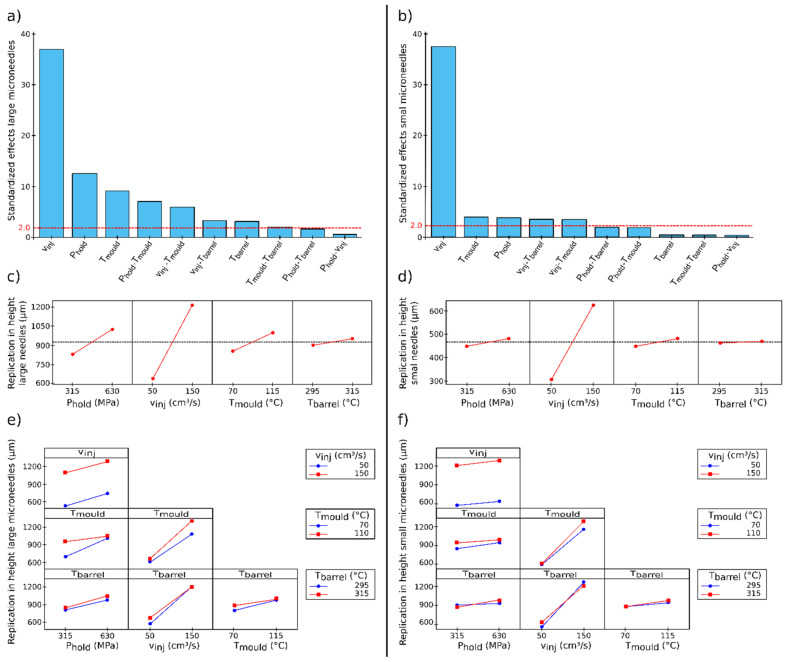
Illustration of the DoE results, with the barrel temperature (T_barrel_), volumetric injection rate (v_inj_), holding pressure (P_hold_) and mould coolant temperature (T_mould_) as the varied parameters. (**a**,**b**) show the Pareto charts for the large and small microneedles, respectively. The red dashed lines represent the significance limit at a confidence level of at 95%. (**c**,**d**) show the main effect plot for the replication in height for the large and small microneedles. (**e**,**f**) show the two-factor interactions for the replication in height for the large and small microneedles, respectively.

**Figure 7 micromachines-13-01280-f007:**
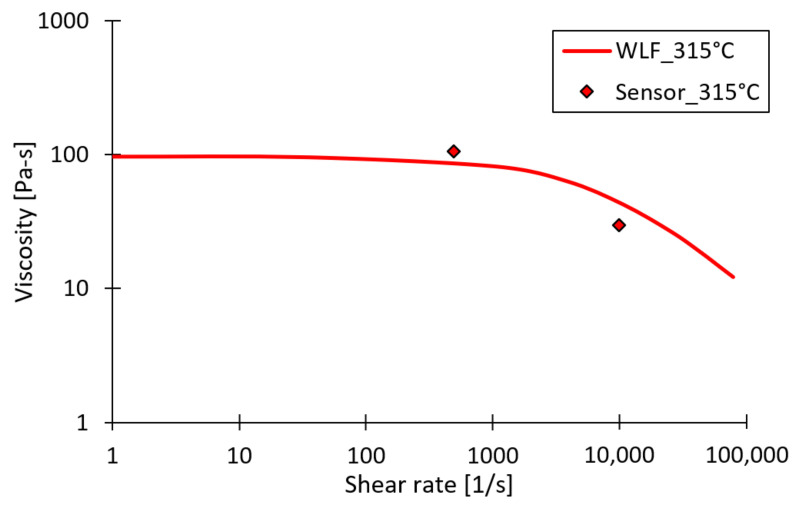
Illustration of the Cross–WLF model viscosity and online apparent viscosity for PC, measured at an injection rate of 50 cm^3^/s and 150 cm^3^/s in combination with a barrel temperature of 315 °C.

**Figure 8 micromachines-13-01280-f008:**
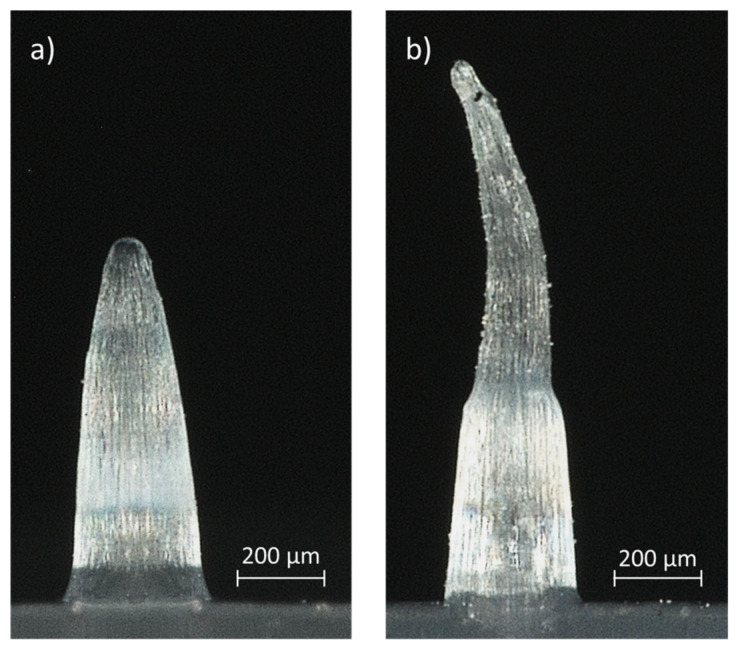
Illustration of a small microneedle which is (**a**) undeformed during the demoulding process, (**b**) deformed during the demoulding process.

**Figure 9 micromachines-13-01280-f009:**
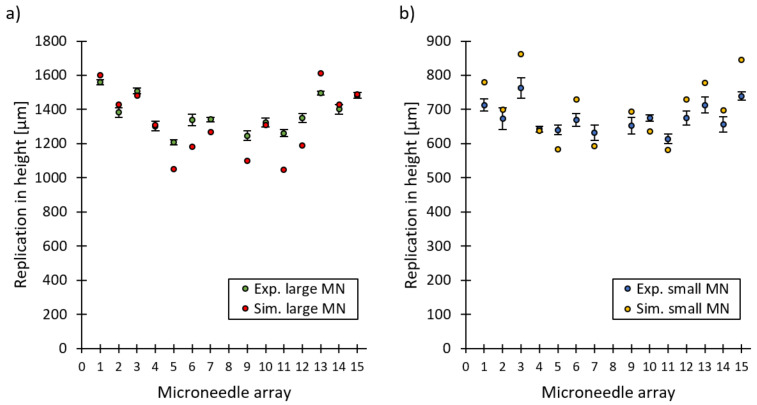
The experimental and simulated replication in height in function of the microneedle arrays, created with the optimal IM parameters for (**a**) the large microneedle arrays and (**b**) the small microneedle arrays. The error bars represent the 95% confidence interval.

**Figure 10 micromachines-13-01280-f010:**
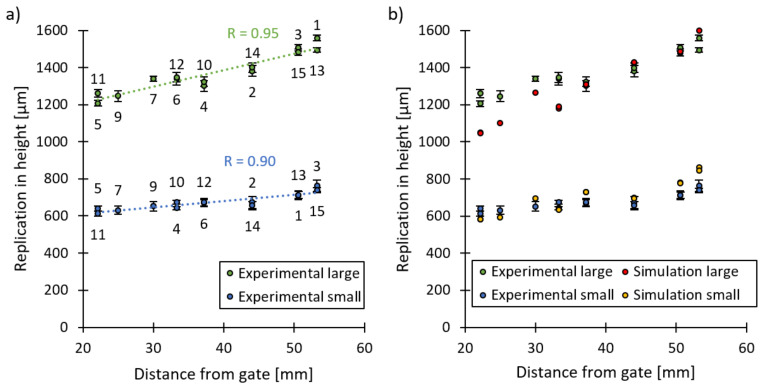
The average replication in height of the microneedle arrays as a function of the distance between the corresponding array and the gate location for (**a**) the large and small microneedle arrays together with a trendline and the linear correlation coefficient (R); (**b**) the experimental and simulated replication in height in function of the distance from the gate. The error bars represent the 95% confidence interval.

**Figure 11 micromachines-13-01280-f011:**
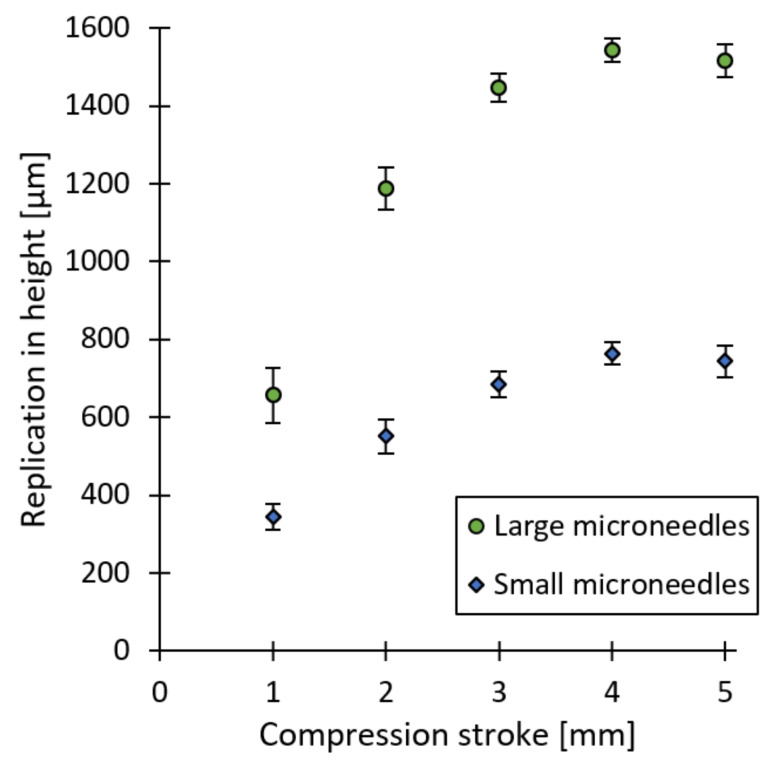
Illustration of the replication in height in function of the compression stroke obtained through ICM, for the large and small microneedles. The error bars represent the 95% confidence interval.

**Figure 12 micromachines-13-01280-f012:**
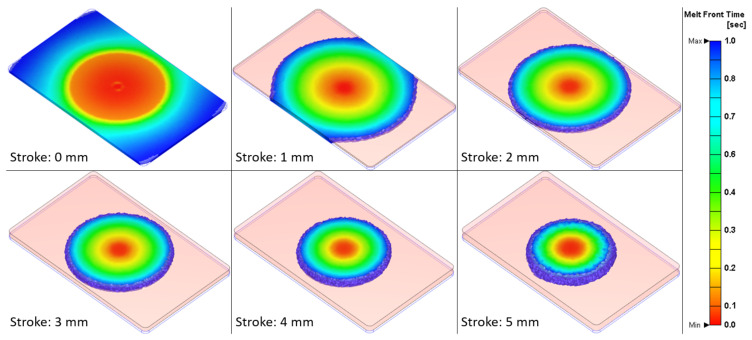
Illustration of the injected melt volume inside the mould cavity, before the mould compression phase, for the different compression strokes. The pink volume represents the compression gap. A compression stroke of 0 mm corresponds to conventional injection moulding.

**Figure 13 micromachines-13-01280-f013:**
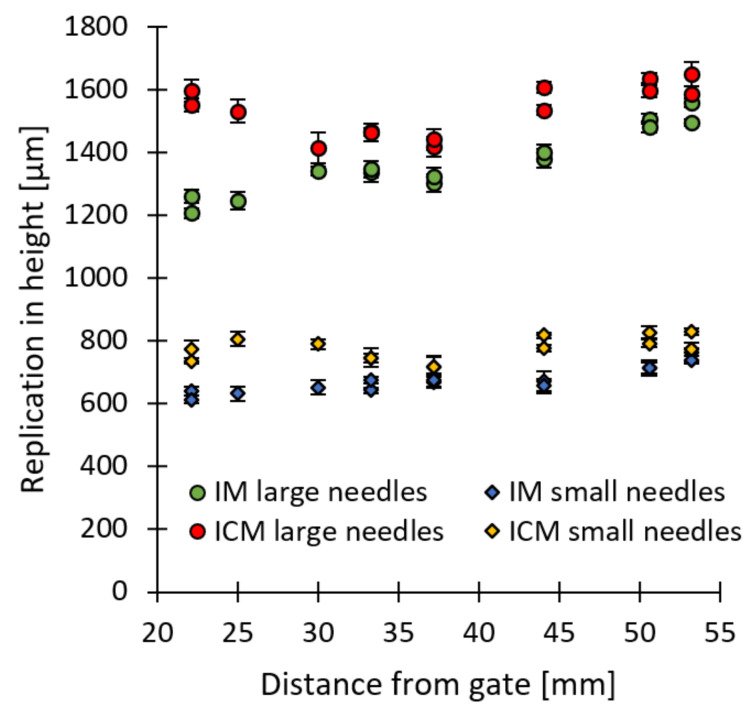
The average replication in height of the microneedle arrays as a function of the distance between the corresponding array and the gate location for injection moulding and injection compression moulding using the process parameters for an optimal replication fidelity.

**Figure 14 micromachines-13-01280-f014:**
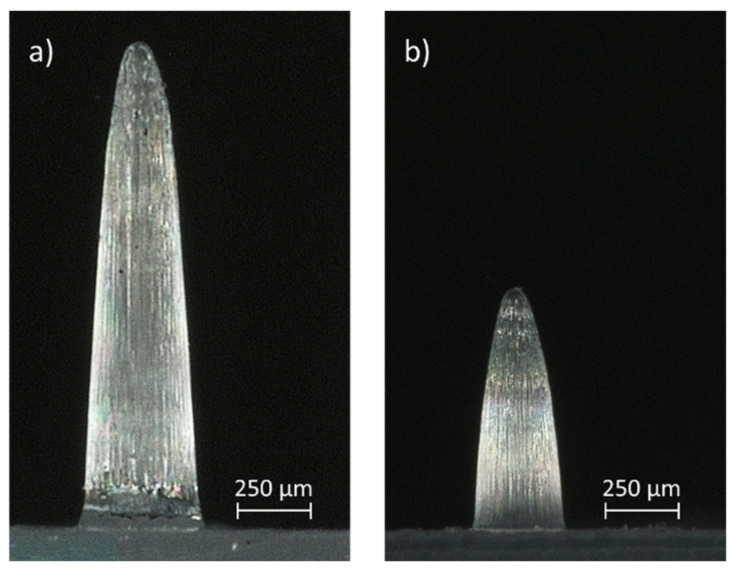
Illustration of a single (**a**) large and (**b**) small microneedle from array 1, created with ICM having a mould stroke of 4 mm.

**Figure 15 micromachines-13-01280-f015:**
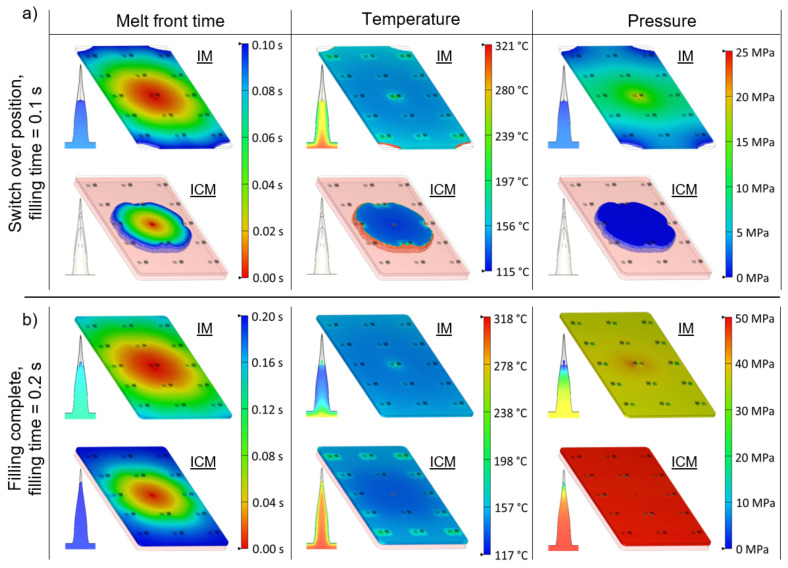
Illustration of the simulated melt front time, temperature and pressure of the macroscopic part and a small microneedle from array 1 for injection moulding (IM) and injection compression moulding (ICM) with a stroke of 4 mm, at (**a**) a filling time of 0.1 s and (**b**) a filling time of 0.2 s.

**Table 1 micromachines-13-01280-t001:** Different settings for the process parameters used within the design of experiments.

DoE Run	Barrel Temperature [°C]	Mould Temperature [°C]	Volumetric Injection Rate [cm^3^/s]	Holding Pressure [MPa]
1	295	70	50	31.5
2	295	70	50	63.0
3	295	70	150	31.5
4	295	70	150	63.0
5	295	115	50	31.5
6	295	115	50	63.0
7	295	115	150	31.5
8	295	115	150	63.0
9	315	70	50	31.5
10	315	70	50	63.0
11	315	70	150	31.5
12	315	70	150	63.0
13	315	115	50	31.5
14	315	115	50	63.0
15	315	115	150	31.5
16	315	115	150	63.0

**Table 2 micromachines-13-01280-t002:** IM simulation parameters.

IM Simulation Parameters	PC
Barrel temperature [°C]	315
Volumetric injection rate [cm^3^/s]	150
Holding pressure [MPa]	63
Mould coolant temperature [°C]	115
HTC during filling stage [W/m^2^∙K]	30,000
HTC packing [W/m^2^∙K]	2500
HTC detached [W/m^2^∙K]	1250
Criterion for stopping calculation	99.9999%

**Table 3 micromachines-13-01280-t003:** IM parameters for an optimal replication fidelity.

IM Parameters	PC
Barrel temperature [°C]	315
Volumetric injection rate [cm^3^/s]	150
Holding pressure [MPa]	63
Mould coolant temperature [°C]	125

## Data Availability

Data underlying the results presented in this paper are not publicly available at this time but may be obtained from the authors upon reasonable request.

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
