# Peer review of "Comparing the Replication Fidelity of Solid Microneedles Using Injection Compression Moulding and Conventional Injection Moulding"

_micromachines, 2022, doi:10.3390/mi13081280_

Round 1
Reviewer 1 Report
This article is interesting and has significant importance for polymer engineering and also medical applications from a practical point of view. It covers both injection molding techniques; conventional injection molding and injection-compression molding, which are widely used in manufacturing industries.
There could be a lot of articles on the effect of process parameters on the replication of micro features with high-aspect ratio. However, the in-depth explanation and analysis on the findings from the molding experiments and CAE simulations make this article distinguishing. The analysis and explanation on the effect of the process parameters of IM and ICM on the replication of microneedles were well described and supported. The authors didn't omit the demolding, deformation, and breakage of the high-aspect ratio microfeatures and well point out the trade-off relationship between the replication and these demolding/ejection issues and future research direction.
The below are my suggestions for better descriptions.
The authors used "barrel temperature" instead of "melt temperature" and it is a more accurate description of the process parameter. However, "mold temperature" was used instead of "coolant temperature". If authors set the coolant temperature to make the measured mold surface temperature from the temperature sensor a specific value, then "mold temperature" is an accurate description. Usually, the measured mold surface temperature is lower than the coolant temperature due to heat loss. The difference between coolant temperature setting and measured mold surface temperature is larger for higher coolant/mold temperature conditions such as >100 degC or insufficient insulation of coolant lines. Therefore, the use of terms "mold temperature" and "coolant temperature" could be critical for the mold temperature setting. Please double check which term is an accurate description among mold and coolant temperature.
* The effect of volumetric injection rate was analyzed by lower viscosity due to the shear heating and shear thinning, and (relative) delay of frozen skin layer formation. Although shear viscosity decreases by shear thinning and heating, a high shear rate can increase melt pressure depending on cavity thickness and shear rate range. According to the experimental setup, a cavity pressure sensor was installed to measure apparent viscosity. Please check the measured cavity pressure from different injection rate conditions. Better replication could also result from increased cavity pressure.
* Please describe the location of cavity pressure and mold temperature sensor.
* Please use "s^-1" or "1/s" instead of "/s".
In my opinion, this well-written article is valuable for readers in a wide range of industries from molding making to high-precision injection molders, also medical applications as well. Therefore, I consider that this article is recommendable to be published in the Micromachines Journal.
Reviewer 2 Report
This paper carried out DoE experimental investigation for the replication fidelity of microneedles using injection compression moulding and conventional injection moulding. Two sizes of microneedles were discussed in this study. Four parameters, i.e., barrel temperature, volumetric injection rate, holding pressure and mould temperature were considered in the DoE investigation for conventional injection moulding. While for injection compression moulding, the injection stroke was also varied and investigated. A comparison between the replication fidelity for the injection moulding and injection compression moulding indicated that the injection compression moulding process resulted in a higher and more uniform replication fidelity.
This paper is well organized with highly logical. The results are convincing and the analysis is detailed. My recommendation is accepted after minor revision.
1. “3.4. Comparison between IM and ICM” should be “3.5. Comparison between IM and ICM”.
2. Add simulation results in the Conclusion.
